# Statistical Machine-Learning Methods for Genomic Prediction Using the SKM Library

**DOI:** 10.3390/genes14051003

**Published:** 2023-04-28

**Authors:** Osval A. Montesinos López, Brandon Alejandro Mosqueda González, Abelardo Montesinos López, José Crossa

**Affiliations:** 1Facultad de Telemática, Universidad de Colima, Colima 28040, Mexico; osval78t@gmail.com; 2Centro de Investigación en Computación (CIC), Instituto Politécnico Nacional (IPN), Mexico City 07738, Mexico; brandon.mosqueda@hotmail.com; 3Centro Universitario de Ciencias Exactas e Ingenierías (CUCEI), Universidad de Guadalajara, Guadalajara 44430, Mexico; 4International Maize and Wheat Improvement Center (CIMMYT), Km 45, Carretera Mexico-Veracruz, El Batan, Texcoco 56237, Estado de Mexico, Mexico; 5Colegio de Postgraduados, Montecillo 56230, Estado de Mexico, Mexico; 6Centre for Crop & Food Innovation, Food Futures Institute, Murdoch University, Murdoch 6150, Australia

**Keywords:** R package, SKM, statistical machine learning, genomic selection

## Abstract

Genomic selection (GS) is revolutionizing plant breeding. However, because it is a predictive methodology, a basic understanding of statistical machine-learning methods is necessary for its successful implementation. This methodology uses a reference population that contains both the phenotypic and genotypic information of genotypes to train a statistical machine-learning method. After optimization, this method is used to make predictions of candidate lines for which only genotypic information is available. However, due to a lack of time and appropriate training, it is difficult for breeders and scientists of related fields to learn all the fundamentals of prediction algorithms. With smart or highly automated software, it is possible for these professionals to appropriately implement any state-of-the-art statistical machine-learning method for its collected data without the need for an exhaustive understanding of statistical machine-learning methods and programing. For this reason, we introduce state-of-the-art statistical machine-learning methods using the Sparse Kernel Methods (SKM) R library, with complete guidelines on how to implement seven statistical machine-learning methods that are available in this library for genomic prediction (*random forest*, *Bayesian models*, *support vector machine*, *gradient boosted machine, generalized linear models*, *partial least squares*, *feed-forward artificial neural networks*). This guide includes details of the functions required to implement each of the methods, as well as others for easily implementing different tuning strategies, cross-validation strategies, and metrics to evaluate the prediction performance and different summary functions that compute it. A toy dataset illustrates how to implement statistical machine-learning methods and facilitate their use by professionals who do not possess a strong background in machine learning and programing.

## 1. Introduction

In the digital era, data abounds everywhere. Each system, each transaction, and each device with which we interact stores information that will later be processed to extract insights and patterns. Data analysis has always been an important topic and is exemplified by the evolution of mathematics and statistics, areas that have allowed us to understand the world. However, since data have been generated in colossal volumes because of digitalization, analyzing them with conventional methods has proved challenging, thus promoting the invention of new disciplines such as machine learning or statistical machine learning. These methods are powerful and can help uncover hidden patterns, extract valuable knowledge, and model complex real-world problems. Statistical machine learning can be used for association studies, prediction studies, density estimation, and dimensionality reduction. However, achieving expertise in these subfields requires extensive training in mathematics and statistics fundamentals.

Because of the continuous growth of knowledge in scientific fields, it is increasingly difficult for experts in one field to become proficient in related fields, such as statistical machine learning, even though these fields can enhance their research. For instance, statistical machine-learning methods for GS are essential for breeders, since GS is a predictive methodology that utilizes a reference population with phenotypic (output) and genotypic (input) data to train a statistical machine-learning method to predict candidate lines with only genotypic information. However, breeders need not possess expertise in statistical machine learning. Instead, they require adequate training in applied and automated tools to extract useful knowledge from their own data.

However, the statistical machine-learning field is constantly evolving, meaning there are regularly new proposals to improve existing algorithms [1,2,3,4] and other application problems [5,6,7,8]. The field of genomic selection is not the exception, and in fact, is a prolific area of study with hundreds of papers published each year where we can find specific algorithms [9,10,11] or new studies that show how other statistical machine-learning methods can predict phenotypic traits with reasonable precision [12,13,14,15]. Xu et al. [16] note that artificial intelligence (AI) can increase the probability of identifying truly favorable genotypes by focusing on current breeding materials, potentially achieving optimal traits. In this sense, AI will continue to develop and create new opportunities for plant breeding.

For this reason, it is necessary to develop software with high levels of automation so that breeders can understand and use it with only a basic training process that focuses on the right use of these tools. For specific areas of knowledge where statistical machine-learning methods are used, multiple programs have been developed [17,18,19] to facilitate the implementation of these algorithms. In practice, when someone has a dataset and wants to evaluate different algorithms that best fit its data, most of the time we would need to invest time in testing different approaches for the problem and looking for the optimal model for the specific dataset. Libraries such as scikit-learn [20] for Python, mlr3 [21] for R, and H2O.ai (H2O.ai, 2016) for Java provide a complete machine-learning toolkit with functions to perform several classic machine-learning tasks such as model fitting, tuning, and evaluation. The Application Programming Interface (API) for each library changes from one app to another, and as they are very complex, it is difficult to use and understand what is happening in the beginning, especially if you are unfamiliar with the paradigm with which they were developed. For example, the mlr3 library uses Object-Oriented Programming (OOP), which might be unfamiliar to those who are not experts in programming. Considering simplicity, speed of implementation, and ease of use, we developed and published the Sparse Kernels Methods (SKM) [22] R library. We believe this library can be a valuable alternative for both beginners in machine learning who seek easy-to-use tools and more experienced users looking for a quick implementation tool without losing the flexibility to control the algorithms available in SKM.

The SKM library is a general-purpose statistical machine-learning library, capable of implementing statistical machine-learning algorithms for any prediction study field when inputs and outputs from a training set exist. Its application for genomic selection is straightforward, as will be illustrated in the following sections, since the SKM library features various automated tools to facilitate its use. As such, we present a brief introduction to the machine-learning workflow and the use of SKM for genomic selection with examples that use the SKM’s functions for fitting models using the seven algorithms, in addition to evaluating their prediction performance.

## 2. Material and Methods

### 2.1. Statistical Machine-Learning Workflow

Sometimes, there is confusion regarding what machine learning involves. Here, we refer to statistical machine learning as inferring a model from data using an algorithm. We do not consider collecting, cleaning, and processing data as part of the machine-learning workflow, as that is part of a bigger concept called data science or data mining. In this sense, when we talk about statistical machine learning, we only refer to learning from data using algorithms. Although learning from data is a broad concept, we are limiting it only to the process of supervised learning for inference, on which the SKM library is focused, so in genomic selection, it is applicable when we have data of a reference population that contains both phenotypic and genotypic information to train a statistical machine-learning method to predict new individuals for which we only have genotypic data. Unsupervised learning and other types of data analysis are not covered here.

We assume we are going to use statistical machine-learning algorithms once we have the data ready to use, preprocessed, and in a suitable format. In this way, the common machine-learning workflow is not just to feed the data to the algorithms but to also select the best hyperparameters that work for our data and evaluate their prediction performance to compare them. With the knowledge there is no universal machine that can give quality results for every problem, we usually use different algorithms and compare their results to select the best one for our problem.

Cross-validation is one of the most widely used methods for algorithm comparison in terms of out-of-sample data. The main goal of cross-validation is to split the data into two mutually exclusive subsets, one for training the algorithm (training set) and the other for evaluating its generalizability (testing set), i.e., how good it is at predicting new samples of the problem not used during training. This method of splitting the data is repeated several times, and in each repetition, we compute prediction errors, dependent on the type of problem we are studying. Finally, we consider the average of these errors as the final error of the algorithm. This process applies to each algorithm we want to test and from here, we decide which algorithm produced the smallest error, the algorithm that best models our problem in terms of prediction performance. There are different types of cross-validation, and they vary based on how the splitting is conducted. For example, k-fold cross-validation consists of dividing the dataset into K mutually independent equal-sized folds. The model is trained on K-1 folds and validated on the remaining fold, repeating the process K times. Finally, the results of each fold are averaged to estimate the model’s generalization performance. Some strategies of cross-validation are more appropriate for specific problems, but essentially, they all work to split the data in training and testing through multiple evaluations.

Cross-validation is also helpful in estimating expected generalization errors for specific scenarios. In genomic selection, we are interested in producing accurate models to predict phenotypic information from genomic data, but these models are also used for predicting genotypes in a complete environment using the genomic information and information of other environments. In general, cross-validation in genomic selection is used to mimic many real plant-breeding goals, including predictions based on tested lines in untested environments, untested lines in untested environments, tested lines in tested environments, and untested lines in tested environments. Normally, in each strategy of cross-validation for each partition (split), we estimate a prediction error without sample data (validation set) with the observed and predicted values, and we report the prediction accuracy as the average prediction error of all partitions.

Another use of cross-validation is for hyperparameter tuning, which refers to the selection of the best hyperparameters that improve the out-of-sample predictions of the algorithm for a particular problem. Usually, a set of hyperparameters is proposed and each hyperparameter combination is evaluated iteratively using some type of cross-validation over the training set to obtain an error measured in the validation set. The combination that gives the smallest error is then selected and used for training one model with the complete training set. Usually, two levels of cross-validation are used in the machine-learning workflow: the first for splitting the data in training and testing to evaluate the algorithm generalizability in out-of-sample data, and the second, an inner cross-validation implemented over the training set to select the best hyperparameters. In general, an iteration of cross-validation can be described as shown in Figure 1.

Application of this procedure to different algorithms will result in one error value associated with each algorithm, allowing a determination of each algorithm’s efficacy. In the following sections, we present examples of how to implement this workflow in the SKM library, focused especially on the hyperparameter tuning and model training phases.

### 2.2. Dataset

To explain the implementation of the machine-learning workflow for genomic selection with the SKM library, we present several examples with a genomic selection dataset in the following sections. This genomic selection dataset comprises 100 lines in five environments (348 observations). The dataset contains the evaluation of 100 lines over a period of five years (2009 to 2013). In addition to this, it includes four continuous traits—grain yield (GY), percentage of head rice recovery (PHR), percentage of chalky grain (GC), and plant height (PH). One of these traits is transformed into a binary and categorical trait with three distinct categories, to demonstrate the application of statistical machine-learning methods available in the SKM library. This diversity in the type of response variables will highlight the versatility of the SKM library. All the source code and data for reproducing the examples presented in this paper can be consulted in the following GitHub repository https://github.com/osval78/SKM_Genomic_Selection_Example (accessed on 25 April 2023). Additional complete examples can also be found in this repository (Appendix A).

### 2.3. SKM Library

SKM is an R library that allows us to implement seven of the most powerful state-of -the-art algorithms (*random forest*, *Bayesian models*, *support vector machine*, *gradient boosted machine, generalized linear models*, *partial least squares*, *feed-forward artificial neural networks*). It does not implement the algorithms itself but internally uses already popular libraries for this purpose, which have efficient and complete implementations of the algorithms. All available algorithms were chosen because they have shown reasonable or good prediction performance in the genomic selection [9,12,15,23], but the benefit of SKM is that it uses a base interface to interact with each algorithm. These were designed considering their specifications of use, which makes learning how to use each library separately unnecessary. In addition, for algorithms with hyperparameters, SKM has an easy way to specify the conditions for hyperparameter tuning, computation of evaluation metrics, automatic creation of cross-validation strategies, and summaries in genomic selection.

### 2.4. Installation

Currently, SKM is not available on the Comprehensive R Archive Network (CRAN) since SKM requires more updated versions of some dependencies such as BGLR and TensorFlow that have yet to be published in this repository. Subsequently, the only option to install it is directly from the repository of GitHub at https://github.com/brandon-mosqueda/SKM (accessed on 25 April 2023). Using the following block of code in the R console, you can install the library from GitHub:



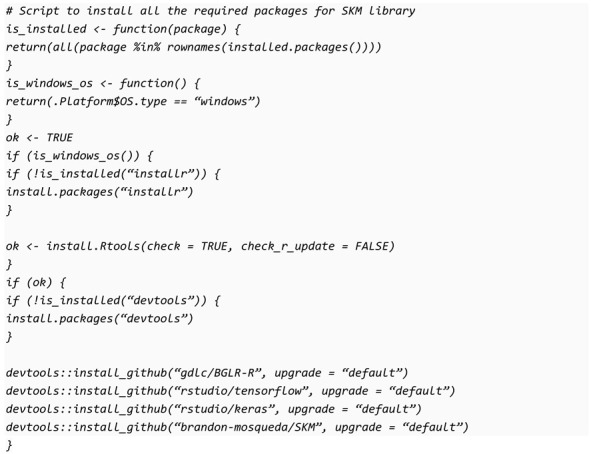



It is important to point out that if you cannot install the devtools with the code above, they are available for download at: https://cran.r-project.org/bin/windows/Rtools/rtools42/files/rtools42-5355-5357.exe (accessed on 25 April 2023) and only you need to execute the .exe file.

### 2.5. Kernels

An important aspect of the SKM library is the availability of kernels to capture nonlinear patterns in data, giving the library its name. We can see a kernel as a function that maps a feature into a high-dimensional space. For this reason, a kernel function is defined as a “similarity” function that corresponds to an inner product in some expanded feature space. This kernel technique is widely used in machine learning because it is easy to implement under a regression/classification problem after the transformation of the input, in a new high-dimensional space.

A great advantage of kernels is that they are compatible with any predictive machine, so we can use any popular kernels with the vast number of options for machine-learning algorithms. As Montesinos-López et al. [23] noted, often we can achieve better predictive results using kernels, primarily when the inputs contain nonlinear patterns, such as many datasets in the context of genomic prediction. In terms of computational efficiency when managing large complex data that show nonlinear patterns, kernels are also excellent options because they reduce the dimensionality of the inputs. In the SKM library, you can find the seven kernels and their sparse versions [23]: Linear, Polynomial, Sigmoid, Gaussian, Exponential, Arc-Cosine 1, and Arc-Cosine L (with L = 2, 3, …). In SKM, it is possible to use the kernelize function, which allows you to easily transform your data using one of the available kernels. Because the kernel computation is independent of the model-fitting process, the kernelize function can be used with any of the seven statistical machine-learning algorithms available in the SKM library. The following block of code shows the interface of this function that returns the kernelized matrix:



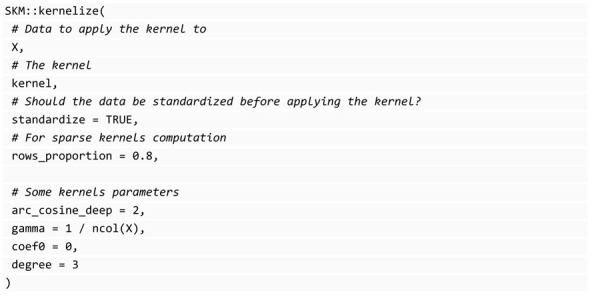



With the help of descriptive examples, we seek to offer a reference material for those interested in using these powerful machine-learning algorithms.

## 3. Results

### 3.1. SKM Workflow for Genomic Selection

When working with R, the first step is to import the required library (only SKM for these examples) and load the dataset used for the analysis, as shown in the following code:



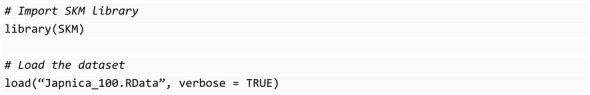



The following subsections include the code required to implement all the steps in the machine-learning workflow described in Figure 1 using the SKM library for genomic prediction.

### 3.2. Data Preparation

Most supervised machine-learning models require the data to be in matrix form, where each row of the matrix represents an observation of the problem, and each column represents an independent (predictor) variable. As they are supervised models, the response (objective) variable must be supplied for each observation in the predictor’s matrix. This is the required format in almost all the algorithms of SKM. As stated above, we do not consider data preparation as part of the machine-learning workflow, but in genomic selection and to clarify how the genomic and phenotypic data are converted into a matrix format for use with the algorithms, we present the most common data preparation in the following block of code:



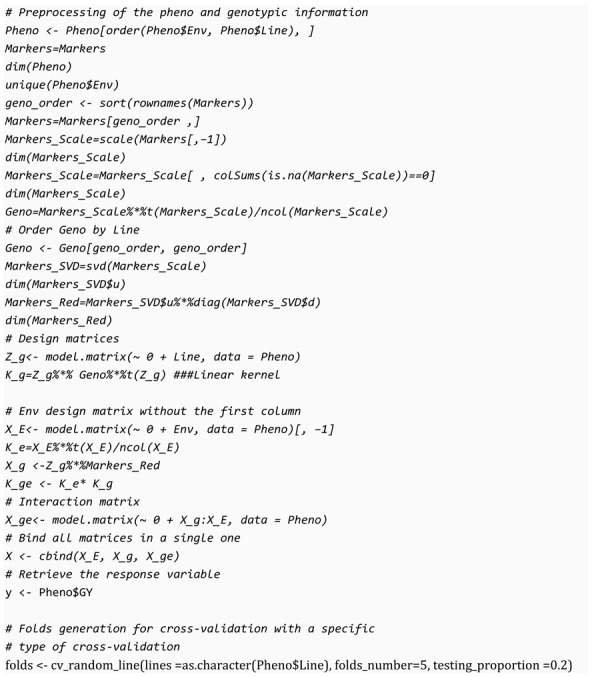



Note the code before was designed to use the variables in the previously described dataset. The first thing to do is sort the phenotypic data by environment and lines, and the genomic data by line. A common approach to genomic selection involves the creation of lines and environment design matrices, where the lines design matrix is post-multiplied by the reduced marker data, which guarantees the same prediction accuracy in using the whole marker data. We can compute an interaction term using this resulting matrix with the environment design matrix and finally, bind these three matrices together, including all the variables going to be used as predictors. In a different vector variable, we hold the response variable denoted by y. Additionally, when the inputs are required in terms of linear or nonlinear kernels with the Hadamark product (element-wise product), the kernels are computed using the interaction terms.

At this point, we generate all the partitions to be evaluated following one method of cross-validation, and as stated before, each method differs from others via the method of splitting. The SKM library includes several functions of cross-validation methods, some of them specifically for genomic selection. The advantage is that we can use any of them indistinctly without affecting the rest of the code; if we want to use a different cross-validation method, we only need to change the line of code where the partitions are created. In the cross-validation section, the available methods are described in detail.

### 3.3. Model Evaluation

With all the data and the partitions ready for cross-validation, we proceed to the evaluation. The following code shows the required steps for machine-learning algorithm evaluation in the genomic selection context:



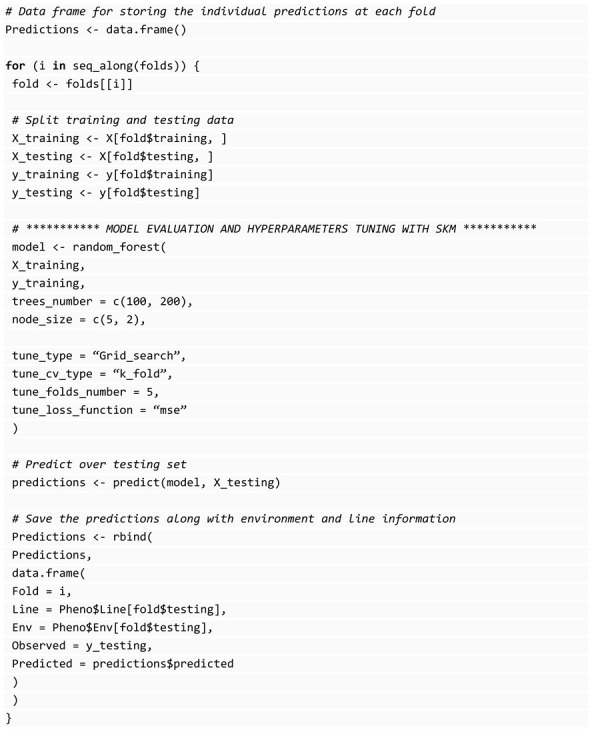



Internally, because SKM infers the type of response variable, it uses the appropriate algorithm configuration to work with each type of data.

We store each prediction in a data frame with information about the partition for which each observation belongs, the observed (true) value, and the environment and line associated with it. We use a for loop to iterate over the generated folds and within the loop; we split our data into training and testing sets, the first one used for hyperparameter tuning and model fitting, and the second one for evaluating the algorithm’s generalizability. Storing predictions in this manner facilitates in-depth analysis of the predictions and as will be demonstrated subsequently, enables the computation of specific summaries of genomic selection, aiding in the identification of the optimal algorithm for our genomic data.

The process of hyperparameter tuning, which also requires cross-validation, is implemented internally by the SKM model functions. These functions include parameters to control the tuning process, such as the hyperparameter combinations to be evaluated and the specifications of internal cross-validation.

### 3.4. Prediction Performance Evaluation

An important aspect of algorithm evaluation is the metric used because there are many ways to measure the model performance in terms of real observed values and the predictions obtained. One way, widely used in regression tasks, is to measure the difference (error) between the observed and predicted values. Another way of measuring model performance is by assessing the degree of agreement commonly used in classification tasks. The main idea of a prediction performance evaluation metric is to compute a single numeric value that captures the performance in terms of error or agreement with which we can compare different predictive machine methods to select the best one. For this reason, in (Table 1) we provide metrics to evaluate the prediction accuracy for regression and classification problems and in one column we indicated the measure type: error-base or agreement-base. Distinguishing between the two measure types is key since when we use an error-based metric, the algorithm with the lowest value is the best in terms of prediction accuracy, while with agreement-based metrics, we select the algorithm that yields the highest value, i.e., the algorithm with the largest values is the best model in terms of prediction accuracy.

The principal reference to selecting one error metric over another is the type of response variable, as some metrics are designed for numeric variables (regression tasks), whereas other metrics are used for categorical variables (classification tasks). We have included in Table 1 all the available metrics in the SKM library for algorithm selection. It is beyond the purpose of this article to present the definition and interpretation of all these metrics and the scenarios where one is more appropriate than the other; however, this table will serve as a reference when working with algorithm selection.

SKM also includes a function for computing the prediction errors in genomic selection, gs_summaries. Unlike all the previously described functions, which work for different areas and expect a vector with the observed values and another with the predicted values, gs_summaries is designed for genomic selection and expects a data frame with five columns: *Fold*, the fold number; *Line*, the line; *Env*, the environment; *Observed* and *Predicted*. The type of response variable is automatically inferred, so numeric or categorical metrics are computed and reported depending on that. This function returns three different summaries computed by line, by environment, and by fold, allowing us to analyze the performance in three different forms that will help us select the best algorithms. The following block of code shows how we can use the function gs_summaries with the predictions in the data frame generated by the example presented in the previous section.



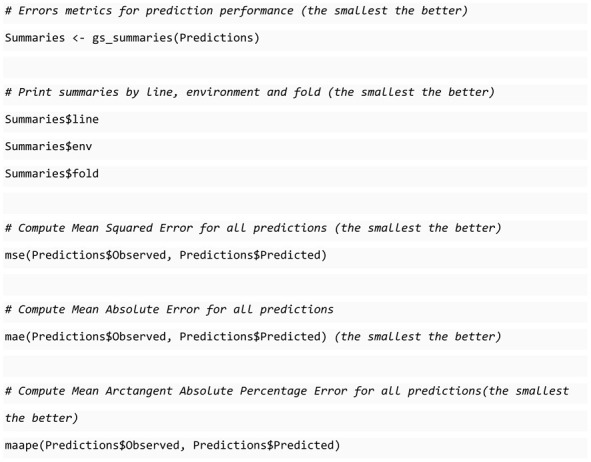



Although the three summaries generated by the gs_summaries are computed for genomic prediction, it is important to describe how they are computed:


**Line summary**


Group the predictions by Line.Observed and predicted values are averaged within each line.The difference between observed and predicted values is computed.Predictions are ordered by the lowest to the greatest difference.


**Environment summary**


Predictions are grouped by Env, Fold, and Line.Observed and predicted values are averaged within each line.Error metrics are computed with the averaged values within each Env-Fold.Error metrics are averaged within each Env and standard error is computed for each metric computed.


**Fold summary**


Predictions are grouped by Fold, Env, and Line.Observed and predicted values are averaged within each line.Error metrics are computed with the averaged values within each Fold-Env.Error metrics are averaged within each fold and standard error is computed for each metric computed.

This description applies to numeric variables. Although the same holds for categorical variables, instead of using the average to summarize the observed and predicted values, the mode (or more frequent value) is used. The environment summary provides insights into the model’s ability to predict data for each specific environment within our dataset, whereas the fold summary assesses the model’s performance in predicting data for each fold, which is valuable when utilizing a specialized form of cross-validation. In contrast, the Line summary evaluates the model’s predictive capabilities for each line independently, regardless of the environment or fold in which the prediction was generated.

### 3.5. Types of Cross-Validation

Cross-validation is another important aspect of machine-learning algorithm evaluation because it allows us to estimate the generalizability expected error of an algorithm in data not used during training. Cross-validation has statistical support and some of its variants for specific areas were proposed to evaluate important study scenarios. For example, in genomic selection, we would want to know how accurate the predictions of a variable measured in an environment not used at all during training are going to be. This scenario can be adapted with cross-validation, and the estimated errors can be interpreted as the expected error of predicting a variable in a completely new environment using information from other environments. The cross-validation methods included in SKM are presented below:**K-fold**: *cv_kfold()* This is the most famous cross-validation method, where k are mutually exclusive subsets of approximately the same size as created. In one iteration, one subset is used as a testing set and the remaining k−1 subsets of data are used for training. This is repeated in such a way that each subset is a testing set at least once, so each observation is guaranteed to be part of some testing set exactly one time.**K stratified**: *cv_kfold_strata()* This is the same as K-fold, except that the subsets are generated considering one categorical variable (usually the response variable) trying to keep the same distribution of the variable in each subset, which prevents an unbalanced distribution of the observations of the smallest groups.**Random partition**: *cv_random()* For this cross-validation method, we need to specify the number of partitions desired and the proportion of observations we want to use as a testing set. In each iteration, a sample of the specified size (without replacement) is drawn from the population to be the testing set and the remaining observations are used as training sets. The disadvantage of this method is that one observation can be a part of more than one testing set or training set.**Random stratified partitions**: *cv_random_strata()* The same as a random partition but with a categorical variable trying to stay balanced in the training and testing sets at each iteration.**Leave-one-group-out**: *cv_leave_one_group_out()* This type of cross-validation is used when we have one variable that exhibits groups in our data. In this way, all data from one group is used as a testing set and data of all the other groups as the training set, so there will be as many iterations as groups in our data. Sometimes it is useful not to predict all the data within a group but only a sample. This method can be adapted to take only a sample subset of data from the testing group and to use all the remaining data as the training set. To be more rigorous, when we are only going to use a sample from the testing group, we must randomly repeat this process several times. Leave-one-group-out is a useful method in a genomic selection where the groups are the environments.**Random line**: *cv_random_line()* This type of cross-validation is specific to the genomic selection context where we have information about lines. It works by taking a sample of the lines in the data, and using all the observations with the selected lines as a testing set. The remaining lines are used as a training set.**Missing**: *cv_na()* When we have data with missing information in the response variable, we cannot use it with supervised learning algorithms, so one thing we can do is to impute the missing variable. However, this is sometimes counterproductive if the imputation method is not efficient because ultimately, we want to generate a model learned from our data to predict new instances of the problem. Another solution is to use these missing observations as the testing set (not for training). Then, we will predict the values but not compute an error metric. Strictly speaking, this should not be considered to be a cross-validation method, but in SKM, this function partitions a dataset in training and testing by its missing values. However, the utility of this function in genomic prediction is that this function can be used for defining the training and testing set in real applications in which we only have phenotypic information for the training set besides independent variables, and for the testing we only have independent variables (markers, environmental information, etc.).

All functions of cross-validation methods return results where each entry is a list with two fields, training, and testing, with the indices of the observations for training and testing sets. As all the functions follow this format, you can easily change to another cross-validation method keeping the rest of the code without modifications.

### 3.6. Hyperparameters Tuning

Hyperparameter tuning consists of evaluating different values in the hyperparameter space in a way that a specific loss function is minimized. Usually, we can set different values for each model’s hyperparameter, and then evaluate all combinations in search of the one that produces the smallest loss value, a method known as grid search. However, this method can be computationally expensive given the large number of combinations required, hence some alternative methods based on probabilistic models, such as Bayesian optimization [24] have been proposed. For tuning the model, we split our training data into (1) inner-training and (2) validation sets. Once the best model is detected by the tuning process, the inner-training and the validation set are jointly used to predict the testing set. Because it happens with any machine-learning algorithms comparison, if we want to be rigorous in the comparison of different hyperparameter combinations, we need to apply cross-validation to reduce the likelihood that the results are a product of chance. The process is repeated using different subsets of the training set, called inner-training and validation sets, to compute the loss value for each combination. In the end, the combination with the smallest loss value is taken as the best combination and used to fit a final model with the entire training set.

Five of the model-fitting functions in SKM are of models with hyperparameters (except for the Bayesian model and partial_least_squares that require hyperparameters but are computed internally efficiently), and these functions require parameters to specify the hyperparameter tuning configuration: tune_type: It can be “Grid_search” or “Bayesian_optimization”.tune_cv_type: Cross-validation for tuning, it can be “K_fold” or “Random” that calls a random partition.tune_folds_number: Number of folds for cross-validation.tune_testing_proportion: For “Random” tune_cv_type, the proportion of individuals for testing (or validation) set.tune_folds: This is a list with the same format used by the cross-validation functions of SKM where you specify the observation’s positions used for inner training and validation. This overrides the specified in tune_cv_type, tune_folds_number, and tune_testing_proportion parameters but allows you to use custom cross-validation for tuning.tune_loss_function: The loss function to optimize during tuning. It is inferred automatically depending on the type of response variable. Mean Squared Error (mse) is used for regression tasks and accuracy for classification, but you can select one specific from the metrics explained in the prediction performance evaluation section.tune_grid_proportion: For “Grid_search” tune_type, the proportion of combinations to evaluate during tuning. By default, all combinations (that is, 1) are evaluated, when you specify a value between 0 and 1, this grid search is equivalent to a random grid search since only a subset of the combinations are evaluated.tune_bayes_samples_number: For “Bayesian_optimization” and denotes the number of initial combinations to evaluate.tune_bayes_iterations_number: For “Bayesian_optimization” and denotes the number of iterations to evaluate after the initial random samples.

It is important to mention the cross-validation performed internally in any of the model’s fitting functions just described only uses the data provided when calling the function, and usually only the training dataset as previously described. In this sense, the training data are split again, into inner training and validation, to perform the cross-validation in search of the best hyperparameter combination.

Depending on the type of tuning selected, the format to provide the hyperparameters changes. For example, for grid search tuning, a vector with the desired values to evaluate is expected in each hyperparameter and all the combinations of the provided values will be evaluated, while for Bayesian optimization, you must provide a list with lower and upper bounds in the hyperparameter space you want to search. The following block of code shows the expected format for hyperparameter tuning for the random-forest algorithm under grid search and Bayesian optimization.



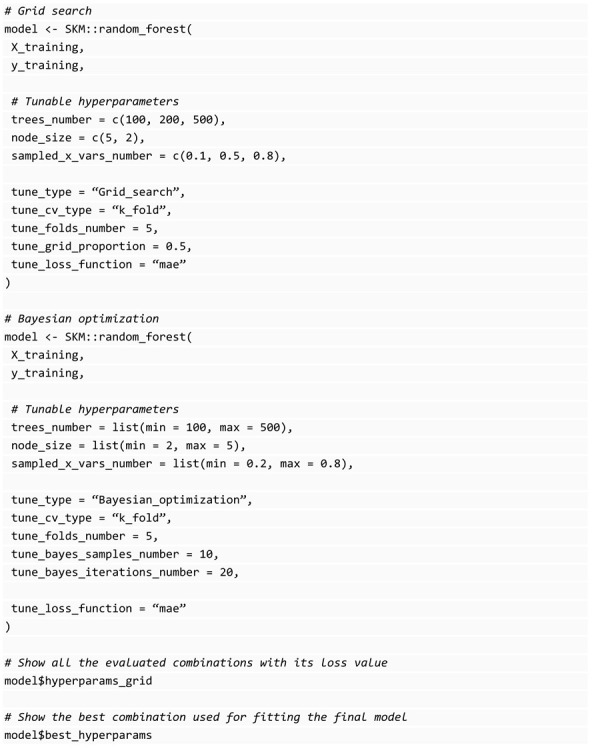



With the grid search example given above, we provide a grid of values for each hyperparameter to be tuned and then internally the resulting combinations (3 × 2 × 3 = 18) are evaluated and the best is selected for the final refitting process of the algorithm, with which are obtained the predictions of the testing set. Although with Bayesian optimization, for each of the hyperparameters to be tuned, a range of values is provided and with this information, this method finds an optimal combination of these hyperparameters that also are used to refit the model with the whole training set, and finally, the predictions of the testing set are performed with this refitted model. 

After selecting the best combination of hyperparameters, the model is refit with the best combination found, using all the provided training datasets. Later, all the evaluated combinations and their respective loss values can be consulted from the model object returned by these functions and finally, the predictions for the testing set are performed with the final refitted model. 

### 3.7. Models in the SKM Library

The model’s functions represent the core of the SKM library as they encapsulate everything necessary for hyperparameter tuning and model fitting, as well as cross-validation methods. The model’s functions share a base interface and can be used interchangeably with one or more models without significantly changing the rest of the code. In this section, the definition of the seven algorithms included in SKM is provided along with code examples.

#### 3.7.1. Random Forest

Random forest (RF) is an extension of bootstrap aggregating, which creates an ensemble of trees and then aggregates their results. Each tree is constructed using a unique splitting criterion (loss function), such as the least-squares criterion for continuous response variables. During training [1], RF randomly selects S bootstrap samples and chooses subsets of features to split the tree nodes. Each tree minimizes the average loss function in the bootstrapped data and follows the algorithm outlined below:For s=1,…,S bootstrap samples $\left\{y_{s},X_{s}\right\}$Step 1. From the original training set, draw bootstrap samples of size $N_{train}$.Step 2. With the bootstrapped data, grow a random-forest tree $T_{s}$ with the appropriate splitting criterion, by recursively repeating the next steps for each terminal node of the tree, until the optimal node size is reached.
-(2a) Randomly draw nf out of the p inputs. nf is a user-specified parameter.-(2b) Pick the best inputs among the nf independent inputs.-(2c) Split the node into two child nodes. The split ends when a stopping criterion is satisfied, such as when a node has less than a predetermined number of observations. No pruning is performed.Step 3. Output the ensemble of trees ${\left\{T_{s}\right\}}_{1}^{S}$.

The predicted value of the testing set $\left(\hat{y_{i}}\right)$ individuals with input $x_{i}$ is calculated as $\hat{y_{i}}=\frac{1}{S}\overset{S}{\underset{s=1}{\sum }}T_{b}\left(x_{i}\right)$ for continuous response variables or using a majority voting for categorical response variables, i.e., ${\hat{y}}_{ic}=majority\;vote\;{\left\{{\hat{C}}_{s}\left(x_{i}\right)\right\}}_{1}^{S}$, where ${\hat{C}}_{s}\left(x_{i}\right)$ is the class prediction of the sth RF tree. For more details on the theory see Breiman [1]. Tree hyperparameters: the number of inputs (features) sampled in each iteration (nf), including the number of trees (ntree); and the number of samples in the final nodes (nodesize) must be defined by the user.

SKM uses the implementation of this algorithm of the library randomForestSRC [25] because it is one of the most complete and efficient ones available. Some parameters were renamed to follow the convention adopted in SKM and to make them more explicit. The SKM version of this algorithm is named random_forest(). Internally, it infers the type of response variable provided and automatically uses the appropriate setting of the algorithm. This function accepts numeric and categorical response variables. In addition, it allows us to fit multivariate models with combinations of both numeric and categorical variables. In the latter case, the response variable is expected to be a data frame, not a vector.

In the following block of code, all the available parameters of the random_forest function in the SKM library and their default values are illustrated.



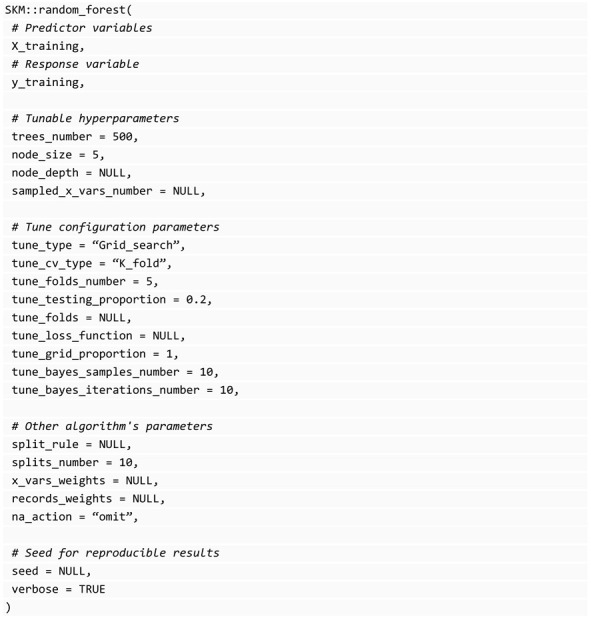



#### 3.7.2. Bayesian Models

In SKM, Bayesian models include all models from the BGLR library [11]. To implement these models using SKM’s bayesian_model() function, the predictor matrix format differs from the other six algorithms in SKM. A list of predictor variable components and their corresponding models must be provided. In addition, all data are used instead of partitioning into training and testing sets. Positions used for testing are set to NA or specified in a designated parameter. This approach is because of the adaptation of the models directly from the BGLR library.

One of the most popular predictors used in genomic prediction under the Bayesian GBLUP model is:$$Y_{ij}=\mu +L_{i}+g_{j}+gL_{ij}+\epsilon_{ij}$$
where $L_{i}$ denotes the random effects of environments (or locations) distributed as $L={\left(L_{1},\ldots ,L_{I}\right)}^{T}∼N_{ℐ}\left(0,\sigma_{L}^{2}H\right)$, where H denotes the environmental relationship matrix computed as proposed by VanRaden (2008), but in place of using genomic information, it is estimated using environmental variables; that is, $H=\frac{X_{L}X_{L}^{T}}{r}$, where $X_{L}$ is the standardize (centered and scaled) matrix of dimension I×r containing the environmental information of I environments and for each environment were measured r environmental covariates; $g_{j},j=1,\ldots ,J$, are the random effects of lines, $gL_{ij}$ are the random effects of environment-line interaction (GE) and $\epsilon_{ij}$ denotes the random error terms assumed normally distributed with mean 0 and variance $\sigma^{2}$. Furthermore, it is assumed that $g={\left(g_{1},\ldots ,g_{J}\right)}^{T}∼N_{J}\left(0,\sigma_{g}^{2}G\right)$, $gL={\left(gL_{11},\ldots ,gL_{1J},\ldots ,gL_{IJ}\right)}^{T}∼N_{ℐJ}\left(0,\sigma_{gL}^{2}H⨀Z_{g}GZ_{g}^{T}\right)$, where G is the genomic relationship matrix [26], ⨀ denotes the Hadamard product and H is the environmental relationship matrix of size I. Of course, it is assumed that you can provide nonlinear kernels to fit this Bayesian model. Additionally, other Bayesian methods are allowed to be implemented under this Bayesian function. 

The following block of code is an example of the implementation of this model with the SKM library, which does not include the tuning configuration parameters because this model does not require hyperparameters. They are provided by default and work well for general situations. The subtle differences between the Bayesian function and other algorithms can be seen in this block of code; however, these differences do not break the compatibility with the other SKM functionalities.



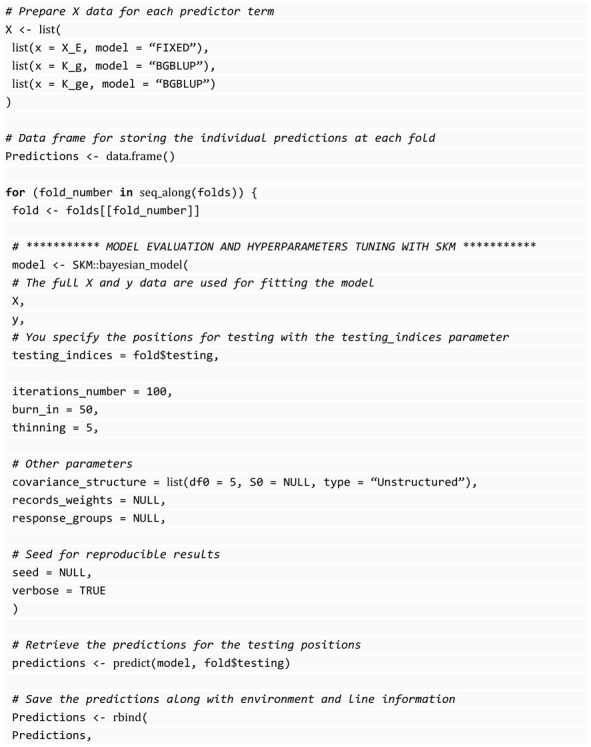





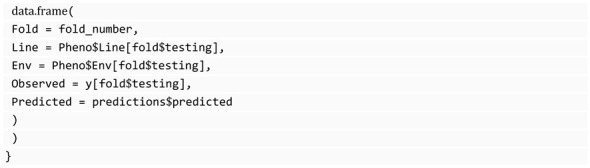



#### 3.7.3. Support Vector Machine (SVM)

SVM is a popular and effective machine-learning algorithm that was introduced to the computer science community in the 1990s primarily for classification problems [27]. The algorithm’s versatility and ability to perform well with a small number of observations make it attractive for solving various problems, including text categorization, speech and image recognition, credit rating analysis, and genomic selection. In essence, SVM is the solution to an optimization problem:$$\mathrm{maximize}:\;\beta_{0},\beta_{11},\beta_{12},\ldots ,\beta_{p1},\beta_{p2},\epsilon_{1},\ldots ,\epsilon_{n}$$
$$\mathrm{subject}\;\mathrm{to}:\;\;\;\;\;\;\;\;\;\overset{p}{\underset{j\;=\;1}{\sum }}\overset{2}{\underset{k\;=\;1}{\sum }}\beta_{jk}^{2}\;=\;1$$
$$y_{i}\left(\beta_{0}+\overset{p}{\underset{j\;=\;1}{\sum }}\beta_{j1}x_{ij}+\overset{p}{\underset{j\;=\;1}{\sum }}\beta_{j2}x_{ij}^{2}\geq M\left(1-\epsilon_{i}\right)\right)$$
$$\epsilon_{i}\geq 0,\overset{n}{\underset{i\;=\;1}{\sum }}\epsilon_{i}\leq T$$
where $\beta_{0},\beta_{11},\beta_{12},\ldots ,\beta_{p1},\beta_{p2}$ are the coefficients of the maximum margin hyperplane. Hyperplanes are reduced subspaces with one less dimension than their original space. The tuning hyperparameter T determines the amount of error allowed, where a smaller T allows for shorter margins and fewer errors. The parameter T is chosen via cross-validation. The margin T is the amplitude we strive to make as large as possible. The error variables $\epsilon_{1},\ldots ,\epsilon_{n}$ allow observations to be on the mistaken side of the hyperplane. The new observation $x^{*}$ is classified based on its position relative to the hyperplane. The classification is based on the sign of the nonlinear function $f\left(x^{*}\right)=\beta_{0}+\beta_{1}x_{1}^{*}+\beta_{2}x_{2}^{*}+\ldots +\beta_{p}x_{p}^{*}$; observations with $f\left(x^{*}\right)>0$ are assigned to class 1 and observations with $f\left(x^{*}\right)<0$ are assigned to class −1. The choice of f(x) f(x) determines the nonlinear function of x to be used.

The support_vector_machine() function in the SKM library uses the e1071 library [28] for training models of this type, for example, an SVM with radial kernel $\left(K\left(x_{i},x_{i^{\prime }}\right)=exp\left(-\gamma \overset{p}{\underset{j=1}{\sum }}\left(x_{ij}-x_{i^{\prime }j}\right)\right)\right)$ with γ as a positive constant that can be implemented using the following code:



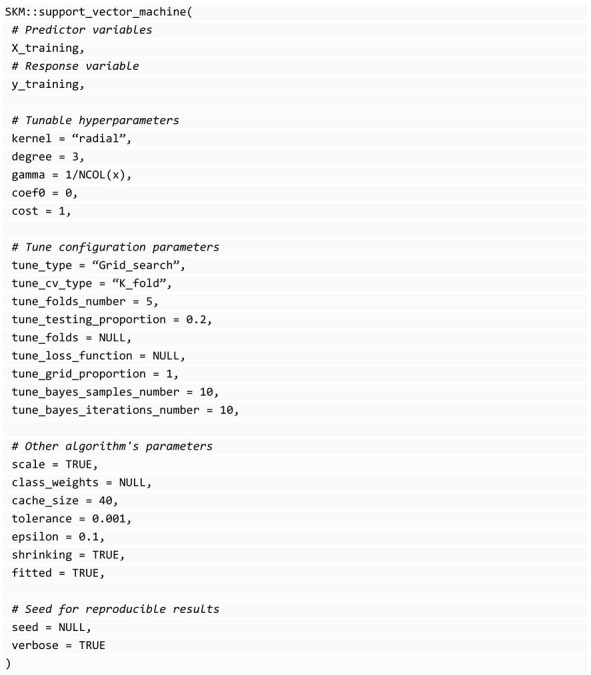



#### 3.7.4. Gradient Boosted Machine (GBM)

The GBM algorithm is a highly effective machine-learning technique used extensively in data-driven applications across fields such as computer science, ecology, biology, and genomic prediction. Its effectiveness has been further enhanced by a modification proposed by Friedman [29] that involves fitting a base learner on a randomly selected subset of the training set, without replacement, at each iteration of the algorithm. This adaptation has demonstrated significant improvements in the prediction performance of gradient boosting. Friedman developed this regression technique based on various fitting criteria and noted its robustness and competitiveness for interpreting regression of unclean data. The GBM algorithm used in our implementation follows the approach proposed by Friedman [29].


**Input**


input data ($y_{i},x_{i}$) for i=1,2,…,nnumber of iterations Mchoice of the loss function ϕ(y,f)choice of the base-learner model h(x,θ)


**Algorithm**

$\mathrm{Step}\;1:\;\mathrm{initialize}\;f_{0}$ with a constant valueStep 2: for t=1 to M repeat Steps 3–6:Step 3: compute the negative gradient of ϕ with respect to $f:g_{t}\left(x_{i}\right)$
$\mathrm{Step}\;4:\;\mathrm{fit}\;a\;\mathrm{new}\;\mathrm{base}-\mathrm{learner}\;\mathrm{function}\;h\left(x,\theta_{t}\right)$ for predicting $g_{t}\left(x_{i}\right)$ from the covariables $x_{i}$
$\mathrm{Step}\;5:\;\mathrm{find}\;\mathrm{the}\;\mathrm{best}\;\mathrm{gradient}\;\mathrm{descent}\;\mathrm{step}-\mathrm{size}\;p_{t}$:

$\;\;\;\;\;\;\;\;p_{t}=arg\underset{\rho }{min}\sum_{i\;=\;1}^{n}\phi \left[y_{i},\hat{f_{t-1}}\left(x_{i}\right)+\rho h\left(x_{i},\theta_{t}\right)\right]$



$\mathrm{Step}\;6:\;\mathrm{update}\;\mathrm{the}\;\mathrm{function}\;\mathrm{estimate}\;\hat{f_{t}}=\hat{f_{t-1}}+\rho h\left(x_{i},\theta_{t}\right)$



$\mathrm{Step}\;7:\;\mathrm{final}\;\mathrm{predictions}:\;\hat{f}\left(x\right)={\hat{f}}_{M}\left(x\right)$




The library gbm of R [30] can be used to implement these kinds of predictive machines and this is what SKM uses internally. As mentioned before, SKM implementation is advantageous because it is a common interface between all the available algorithms and the parameters for automatic hyperparameter tuning. The following block of code provides the specification of the generalized_boosted_machine() function:



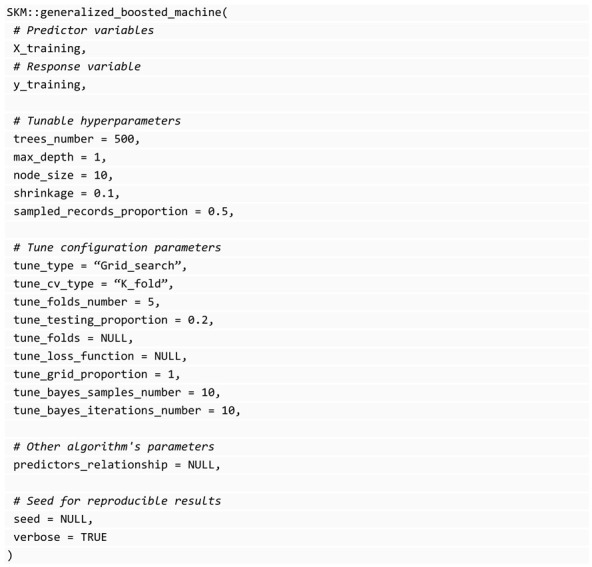



#### 3.7.5. Generalized Linear Models (GLM)

Under generalized linear models, we use a set of p predictor variables and a response variable to fit a model of the form:

*Distribution*: $y_{i}$ is distributed among those which are considered exponential families of probability distributions,
$$\mathrm{Linear}\mathrm{predictor}:\;\eta_{i}=\beta_{0}+X_{i1}\beta_{1}+\cdots +X_{ip}\beta_{p}$$
$$\mathrm{Link}\mathrm{function}:\;\mu_{i}=\eta_{i}$$
where $y_{i}$ is the response variable for the i-th individual (sample), $x_{i}$ is the i− th predictor variable, $\beta_{0}$ denotes and intercept term, $\beta_{i}$ is the beta coefficient corresponding to the predictor $x_{i}.$ The loss function that is minimized is: $$minimize\{-\ell \left(\beta ;y\right)+0.5\lambda \overset{P}{\underset{j=1}{\sum }}\left\{\left(1-\alpha \right)\beta_{j}^{2}+\alpha \left|\beta_{j}\right|\right\}$$
where λ is the penalization parameter, when α = 0 a ridge penalization is implemented, while when α = 1 a lasso penalization is implemented, and when 0 < α< 1 the elastic net penalization is implemented. Since this model is under a generalized linear framework, prediction models for continuous (identity link function with response variable assuming a normal distribution), binary (logit link function with response variable assuming a binomial distribution), categorical (generalized logit link function with response variable assuming a multinomial distribution) and counts (exponential logit link function with response variable assuming a Poisson distribution) response variables can be implemented. To implement these kinds of models, we can use the glmnet [31] library for R that contains efficient implementations. This library is what SKM uses internally in the generalized_linear_model() function. Under the glmnet library for continuous response variables, it is possible to train multi-trait models, which are inherited also by the function generalized_linear_model(). The available parameters for this function are shown in the following block of code:



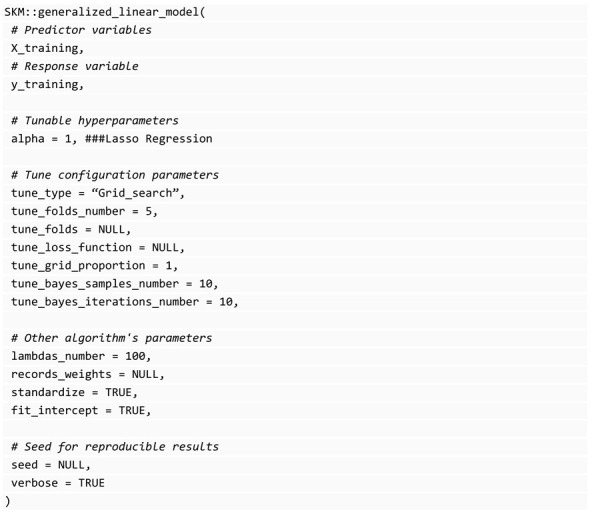



#### 3.7.6. Partial Least Squares (PLS)

PLS regression was first introduced by Wold [32] and was originally developed for econometrics and chemometrics. It is a multivariate statistical technique designed to deal with the p>n problem, i.e., when the number of explanatory variables (p) is much larger (and more highly correlated) than the number of observations (n). The PLS works for relating one or more response variables (Y) to a set of explanatory variables (X). For PLS regression problems, the so-called Latent Variables (LVs) are obtained under an iterative process. One starts with the SVD of the cross-product matrix $S=X^{T}Y$, therefore including information on the variation in both X and Y, and on the correlation between them. The first right and left singular vectors q and w provides the weights vectors for Y and X, respectively, to obtain scores t and u:y=Xw=Ew
u=Yq=Fq
where E and F are initialized as both X and Y, respectively. The X scores t are often normalized:$$t=\frac{t}{\sqrt{t^{T}t}}$$

The Y scores u are not necessary for the regression but are often saved for interpretation purposes. Next, Y and X loadings are computed by regressing against the same vector t:$$p=E^{T}t$$
$$q=F^{T}t$$

Finally, the data matrices are “deflated”: the information associated with this latent variable, in terms of outer products $tp^{T}$ and $tq^{T}$, are subtracted from the (current) matrices E and F.
$$E_{n+1}=E_{n}-tp^{T}$$
$$F_{n+1}=F_{n}-tq^{T}$$

The estimation of the next component can then start from the SVD of the cross-product matrix $E_{n+1}F_{n+1}$. After each iteration, vectors w, t, p, and q are used as columns in matrices W, T, P, and Q, respectively. One difficulty is that columns of matrix W cannot be benchmarked directly: they are estimated from consecutive deflated matrices E and F. It has been shown that an alternative way to represent the weights is that all columns relate to the original X matrix given by
$$R=W{\left(P^{T}W\right)}^{-1}$$

Now, instead of regressing Y on X, we use T scores to calculate the regression coefficients, and later convert these back to the realm of the original variables by pre-multiplying with matrix R (since T=XR):$$B=Rb=RQ^{T}$$
with $b={\left(T^{T}T\right)}^{-1}T^{T}Y$. Only the first *a* components are used for the estimation of b. Since regression and dimension reduction are performed simultaneously, B, T, W, P, and Q are all part of the output. Both X and Y are considered when calculating the LV in T. Moreover, they are defined so that the covariance between the LV and the response is maximized. Finally, predictions for new data ($X_{new}$) should be calculated as:$$Y_{new}=X_{new}B=X_{new}Rb=T_{new}b$$
with $T_{new}=X_{new}R$. The number of optimal components must be determined, usually by cross-validation. We used the root mean squared error of prediction (RMSEP), which was minimized with 10-fold cross-validation in the training dataset and for each value of LV [33].

The partial_least_squares function in SKM provides a wrapper for the pls [34] library that uses 10-fold cross-validation to select the optimal number of components. This algorithm only computes the optimal number of principal components as a tuning parameter, but since this works well, this is not tuned. An example of use is provided below:



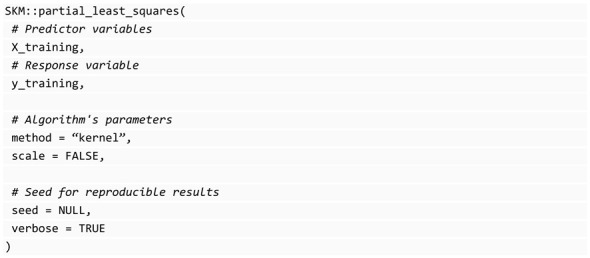



#### 3.7.7. Feed-Forward Artificial Neural Networks

The SKM library provides the capability to implement feed-forward artificial neural networks, specifically multilayer perceptron networks, with the use of the Keras library [35]. The network structure does not assume any specific pattern in the input features. The architecture of a densely connected network comprises an input layer, an output layer, and multiple hidden layers in between. The input layer receives all the independent variables designed to be associated with the output. The net input of each neuron in the first hidden layer is a weighted sum of the independent variables and their corresponding weights, along with an intercept, to which a nonlinear transformation (activation function) applies to capture complex patterns. The output of the first hidden layer is used as input for the neurons of the second hidden layer, and so on. In the output layer, for regression univariate analysis, a one-neuron layer with a suitable activation function is used depending on the type of response variable. Artificial neural networks are known for their various hyperparameters, and SKM can tune them while still allowing the implementation of feed-forward architectures. However, a complex tuning process is required for a successful implementation of artificial neural networks. An example code is provided in the SKM library for implementing a feed-forward network with one hidden layer. The output layer is automatically assigned based on the type of response variable specified in the SKM library; see Montesinos-López et al. [23] for more details about artificial neural networks.



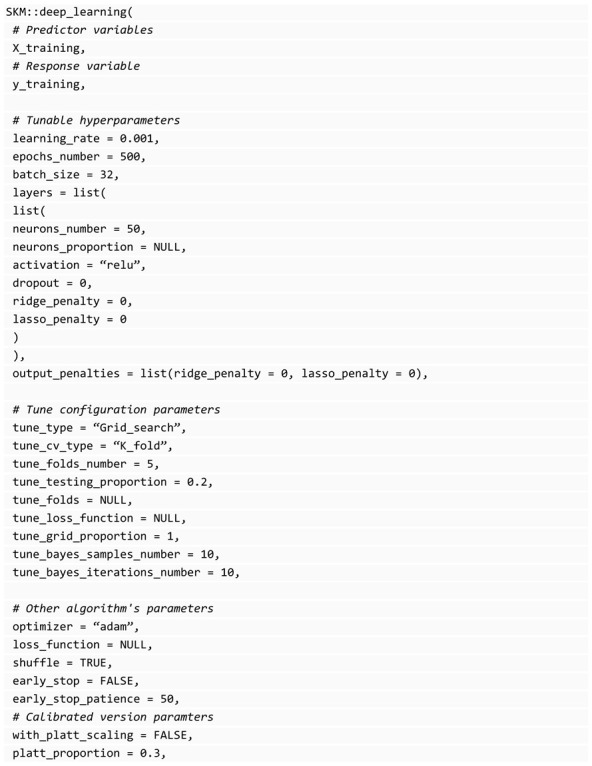





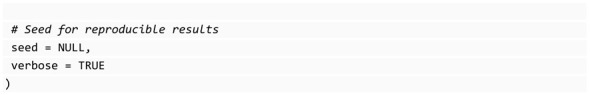



## 4. Discussion

GS requires the successful implementation of statistical machine-learning methods. However, breeders and other professionals without a strong background in statistics and programming may face challenges in accessing software with the necessary level of automation to implement these state-of-the-art methods. To address this issue, we have introduced the SKM library for the R programming language, a flexible and user-friendly tool that enables the implementation of supervised prediction models with minimal coding. The code examples provided in this article and the comprehensive documentation of the package facilitate the application of this powerful learning algorithm across a broad range of data-based problems, particularly suited for genomic selection.

The focus of this paper is to provide detailed guidance on implementing the seven supervised statistical machine-learning methods available in the SKM library (*random forest*, *Bayesian models*, *support vector machine*, *gradient boosted machine, generalized linear models*, *partial least squares*, *feed-forward artificial neural networks*). It is worth noting this library can be utilized by any user, regardless of their level of expertise in statistical machine learning or programming, because (1) six of the seven algorithms available in the library follow the same input format and function implementation format, making it easy to use; (2) it provides numerous options for cross-validation strategies that are simple to implement; (3) it offers state-of-the-art alternatives for parameter tuning that have a consistent format across all algorithms; (4) it offers various metrics for computing the accuracy of the resulting predictions, which are straightforward to use and automatically detect the type of response variable to compute appropriate metrics; (5) it provides comprehensive information on genomic prediction summary metrics for evaluating prediction accuracy in the testing set; (6) it is open access and highly automated, making it user-friendly; and (7) the seven supervised statistical machine-learning methods can be employed with various kernel methods.

To utilize the SKM library effectively, users only require basic knowledge of R programming and supervised statistical machine-learning methods such as understanding the types of data (continuous, binary, categorical, count), basic knowledge of models (pros and cons, inputs and outputs, the ability to capture linear/nonlinear patterns, etc.), types of kernels, preprocessing tools for supervised methods, knowledge of overfitting/underfitting, cross-validation strategies, the importance of tuning process and existing strategies, metrics for evaluating prediction performance, and interpretation of outputs. 

SKM library utilizes a range of cutting-edge prediction algorithms and kernel techniques. The seven kernel techniques available in the library include Linear, Polynomial, Sigmoid, Gaussian, Exponential, Arc-Cosine 1, and Arc-Cosine L, where L can be equal to 2, 3, or other values. These kernel techniques are proficient in identifying nonlinear patterns in input data, which can significantly enhance the accuracy of predictions when such patterns exist. For instance, in a classification problem, a kernel can map the input data to a higher-dimensional feature space where the data becomes more separable and easier to classify. Similarly, in a regression problem, a kernel can capture nonlinear relationships between the input variables and the output variable, leading to more accurate predictions. Please note that kernels are mathematical functions that transform data from one space to another, which can help identify patterns in the data that may not be apparent in its original form. Using kernels can therefore be crucial in improving prediction accuracy, particularly in cases where input data are not linear [36]. For this reason, the SKM library, which allows the implementation of any of the seven kernels with any of the seven state-of-the-art statistical machine-learning methods, is very attractive for genomic selection.

In the Appendix A (or link: https://github.com/osval78/SKM_Genomic_Selection_Example (accessed on 25 April 2023)) we provide six complete examples of the SKM library for genomic selection, which differ only in the type of output data used, the algorithm, the type of cross-validation, type of kernels used, the type of tuning implemented and if the prediction problem is uni-trait or multi-trait. For these reasons, these examples have almost identical codes. Additionally, these codes compute the metrics for evaluating the prediction accuracy of each dataset straightforwardly since we only need to use the summaries function that automatically computes the appropriate metrics depending on the type of response variable, if the response variable is continuous compute metrics such as MSE, NRMSE, Pearson’s correlation, etc., while if the response variable is categorical metrics such as Accuracy, Precision, Recall, Sensitivity, Specificity, Kappa coefficient, etc. are computed. The code from these examples can be used by new practitioners to apply the SKM library to their own data with small modifications. 

The SKM library did not include ensemble learning, which is a machine-learning technique that involves combining multiple models to improve the accuracy and robustness of the predictions. Ensemble learning has many advantages, and some disadvantages, including: (1) increased complexity by combining multiple models, which can increase the complexity of the overall system. This can make it harder to debug and maintain the system; (2) increased computational resources as it requires additional computational resources to train and run multiple models. This can make it more expensive to implement in production environments; (3) can cause overfitting leading to poor generalization performance on new data; (4) might reduce interpretability; (4) can make it harder to interpret the underlying logic behind the predictions and thus making difficult to understand how the model is making its decisions; (5) dependent on the performance of the base models If the base models are not accurate, then the ensemble model will also be inaccurate.

One crucial goal of the SKM library is to provide user-friendly software for genomic prediction, enabling researchers and practitioners to analyze easily and efficiently large and complex genomic data and make accurate predictions about various traits of interest. This is important because advanced statistical and computational methods are often required to analyze genomic data effectively, which can be a significant barrier for those without expertise in these areas. Friendly software for genomic prediction helps to overcome these barriers by providing user-friendly interfaces and workflows that simplify the process of analyzing genomic data, allowing researchers and practitioners to focus on the biological questions they want to answer. Additionally, friendly software for genomic prediction can facilitate sharing and replication of research findings, as others can easily reproduce analyses and results. In summary, friendly software for genomic prediction is essential for advancing our understanding of genetics and genomics and for developing practical applications of this knowledge in fields such as agriculture, medicine, and conservation.

Although the SKM library provides some tools for genomic selection, many other bioinformatics tools are useful for crop improvement and are not covered in the SKM library. Bioinformatics plays a crucial role in crop improvement by providing tools and insights that can be used to develop crops with improved traits and increased yields. In general, bioinformatics has become an essential tool for crop improvement as it allows for the analysis and interpretation of large amounts of genetic and genomic data generated from crops. For example, some bioinformatics tools are used to design guide RNAs for genome editing techniques such as CRISPR/Cas9, which allow for precise modifications to be made to the crop’s DNA and can lead to the development of crops with improved traits.

## 5. Conclusions

This publication aims to encourage the adoption of the SKM library, a versatile statistical machine-learning library that offers a range of utilities for genomic selection applications. In this paper, we demonstrate how the library can be leveraged to implement seven innovative prediction algorithms with minimal coding, thanks to its inherent level of automation (*random forest*, *Bayesian models*, *support vector machine*, *gradient boosted machine, generalized linear models*, *partial least squares*, *feed-forward artificial neural networks*). The implementation format for the seven statistical machine-learning algorithms available in the library is almost identical, making it easy for users to learn and apply these algorithms. Users can specify the type of loss function they prefer by simply indicating the response variable type. The library also offers several cross-validation strategies that are useful in the context of plant-breeding programs. Regarding prediction accuracy metrics, the library provides numerous options, and by default includes several appropriate metrics based on the type of response variables to be predicted. Hyperparameter tuning strategies for each algorithm are readily available, simplifying the implementation of these state-of-the-art statistical machine-learning methods. Despite not being fully automated, the library is highly flexible and can be utilized beyond the context of genomic prediction.

## Figures and Tables

**Figure 1 genes-14-01003-f001:**
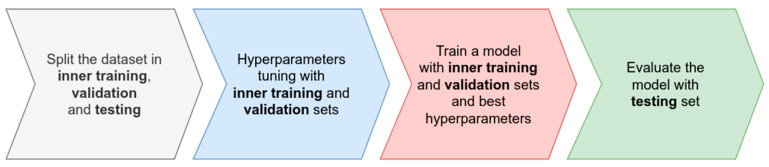
Machine-learning algorithm evaluation in one iteration of cross-validation. Four basic steps are followed. Initially, data are split into inner training, validation, and testing. Then, the is the hyperparameters are tuned using inner training and validation. An important step is the training of the model using inner training, and the validation set using the best hyperparameters obtained in the tuning stage. Finally, the evaluation of the model with the testing set is performed.

**Table 1 genes-14-01003-t001:** Metrics for evaluating model performance in both regression and classification problems. The metrics available in the SKM library are used for this purpose. Numeric metrics are used for regression problems and categorical metrics are used for classification problems. Additionally, the sentence explains that error metrics are useful for regression problems, while agreement metrics are useful for classification problems.

Numeric for Regression Problems		Category for Classification Problems	
Metric	Measure Type	Metric	Measure Type
Mean Squared Error (mse)	error	Accuracy (accuracy)	agreement
Root Mean Squared Error (rmse)	error	Precision (precision)	agreement
Normalized Root Mean Squared Error (nrmse)	error	Recall (recall)	agreement
Mean Absolute Error (mae)	error	Sensitivity (sensitivity),	agreement
Mean Arctangent Absolute Percentage Error (maape)	error	Specificity (specificity),	agreement
Pearson’s correlation (pearson)	agreement	Brier’s score (brier_score),	error
*R*^2^ (r2)	agreement	F1 score (f1_score),	agreement
----	---	Kappa coefficient (kappa_coeff),	agreement
---	---	Matthews coefficient (matthews_coeff),	agreement
---	---	ROC area under the curve (roc_auc)	agreement
---	---	Precision-recall area under curve (pr_auc)	agreement

## Data Availability

Not applicable.

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
