# Peer review of "Statistical Machine-Learning Methods for Genomic Prediction Using the SKM Library"

_genes, 2023, doi:10.3390/genes14051003_

Round 1

Reviewer 1 Report

This study developed a statistical machine learning methods called SKM library to assist genomic prediction and this method provides several different model trainings that are easily for users to follow. However, there are some concerns below need to be addressed before further consideration.

Major concern

This study provides a detailed tutorial to guide users to perform the SKM. However, there is rare result showing comparison of algorithm performance within this library. Although the aim of the SKM library is to aggregate different methods to provide a more friendly way to help users conduct training and prediction, the authors need to provide exact examples including performance of each method and may give some advice for users to select the models given some specific conditions. Also, have authors considered the ensemble learning to improve the performance?

Other concerns

On page 3, for the introduction of cross-validation, the authors may mention a little bit relating x-fold validation other than in the result part.

On page 4 the dataset description is better to summarize as a table that records references, links, etc.

On page 4 for SKM library, the seven algorithms need to be mentioned around here.

Author Response

REVIEWER 1

Open Review

Quality of English Language

( ) English very difficult to understand/incomprehensible
( ) Extensive editing of English language and style required
( ) Moderate English changes required
( ) English language and style are fine/minor spell check required
(x) I am not qualified to assess the quality of English in this paper

Yes

Can be improved

Must be improved

Not applicable

Does the introduction provide sufficient background and include all relevant references?

( )

(x)

( )

( )

Are all the cited references relevant to the research?

(x)

( )

( )

( )

Is the research design appropriate?

( )

(x)

( )

( )

Are the methods adequately described?

(x)

( )

( )

( )

Are the results clearly presented?

( )

(x)

( )

( )

Are the conclusions supported by the results?

( )

(x)

( )

( )

Comments and Suggestions for Authors

This study developed a statistical machine learning methods called SKM library to assist genomic prediction and this method provides several different model trainings that are easily for users to follow. However, there are some concerns below need to be addressed before further consideration.

Major concern

This study provides a detailed tutorial to guide users to perform the SKM. However, there is rare result showing comparison of algorithm performance within this library. Although the aim of the SKM library is to aggregate different methods to provide a more friendly way to help users conduct training and prediction, the authors need to provide exact examples including performance of each method and may give some advice for users to select the models given some specific conditions. Also, have authors considered the ensemble learning to improve the performance?

RESPONSE: Thanks for your time revising this article. With regard to ensemble it was not considered, but regarding to the examples the data and code for implementing each of the seven algorithms these are provided in the following link: https://github.com/osval78/SKM_Genomic_Selection_Example See lines 131-137

Also see citation of Supplementary Material at lines 1024-1034 with details on the examples used.

Other concerns

On page 3, for the introduction of cross-validation, the authors may mention a little bit relating x-fold validation other than in the result part.

RESPONSE: Correction done, and addition made in the new version of the paper. See lines 99-103.

On page 4 the dataset description is better to summarize as a table that records references, links, etc.

RESPONSE: We have considered your suggestion. See lines 131-137.

On page 4 for SKM library, the seven algorithms need to be mentioned around here.

RESPONSE Yes indeed. Many thanks.  See lines 140-141.

Reviewer 2 Report

Top of Form

Genomic-based selection has widened its applications for crop improvement and plant breeding. Authors have used this method to predict candidate genes and introduced new state-of-the-art statistical machine learning methods using the Sparse Kernel Methods (SKM) R library, with guidelines on how to implement seven statistical machine learning methods that are available in this library for genomic prediction. Bottom of Form

The study is highly useful for researchers in plant science and to develop machine learning approaches for crop enhancement.

Few suggestions:

Authors could give attention to latest alternative methods like use of bioinformatics for crop improvement.

Figure 1. Please describe the caption for this figure in details.

The R scripts could be included as supplementary data.

Table 1 could be better reorganized displaying commonalities and differences.

Coloring of text could have been avoided. What is its purpose?

Discussion needs to elaborated, it is too short and abrupt.

After revision, the manuscript could be considered for publication.

Top of Form

Author Response

REVIEWER 2

Open Review

Quality of English Language

( ) English very difficult to understand/incomprehensible
( ) Extensive editing of English language and style required
( ) Moderate English changes required
(x) English language and style are fine/minor spell check required
( ) I am not qualified to assess the quality of English in this paper

Yes

Can be improved

Must be improved

Not applicable

Does the introduction provide sufficient background and include all relevant references?

( )

(x)

( )

( )

Are all the cited references relevant to the research?

( )

(x)

( )

( )

Is the research design appropriate?

(x)

( )

( )

( )

Are the methods adequately described?

( )

(x)

( )

( )

Are the results clearly presented?

( )

(x)

( )

( )

Are the conclusions supported by the results?

( )

(x)

( )

( )

Comments and Suggestions for Authors

Genomic-based selection has widened its applications for crop improvement and plant breeding. Authors have used this method to predict candidate genes and introduced new state-of-the-art statistical machine learning methods using the Sparse Kernel Methods (SKM) R library, with guidelines on how to implement seven statistical machine learning methods that are available in this library for genomic prediction.

The study is highly useful for researchers in plant science and to develop machine learning approaches for crop enhancement.

RESPONSE: Many thanks for the encouragement and positive comments.

Few suggestions:

Authors could give attention to latest alternative methods like use of bioinformatics for crop improvement.

RESPONSE: Correction done in the new version of the paper. We included other bioinformatics tools useful for crop improvements See added comments on important bioinformatics tools at lines 1040-1046.

Figure 1. Please describe the caption for this figure in details.

RESPONSE: Correction done in the new version of the paper. See lines 123-127 for Figure 1. Also see the description of the header of Table 1 (at line 366)

The R scripts could be included as supplementary data.

RESPONSE: All R script and data used are given in . All the source code and data for reproducing the examples presented in this paper can be consulted in the following Github repository https://github.com/osval78/SKM_Genomic_Selection_Example. Additional complete examples can also be found in this repositoryhttps://github.com/osval78/SKM_Genomic_Selection_Example. See lines 140-142

Table 1 could be better reorganized displaying commonalities and differences.

RESPONSE: Header of Table 1 was corrected and completed

Coloring of text could have been avoided. What is its purpose?

RESPONSE: Correction done in the new version of the paper. All colors were removed from the text.

Discussion needs to elaborated, it is too short and abrupt.

RESPONSE: Yes, and thanks. At the Discussion section please see lines 1012-1050

After revision, the manuscript could be considered for publication.

RESPONSE: Thanks for your time.  All suggestions incorporated.

Round 2

Reviewer 1 Report

Thanks for addressed most of concerns. But one thing the author needs to address a potential reason of not implementing ensemble learning.  

Author Response

REVIEWER 1 ROUND 2

Comments and Suggestions for Authors

Thanks for addressed most of concerns. But one thing the author needs to address a potential reason of not implementing ensemble learning.  

RESPONSE: We have added at the DISCUSSIONS possible disadvantages of ensemble learning might. See lines 1035-1044
